# The Roles of Non-Coding RNAs in the Pathogenesis of Uterine Fibroids

**DOI:** 10.3390/cells14161290

**Published:** 2025-08-20

**Authors:** Drake Boos, Tsai-Der Chuang, Omid Khorram

**Affiliations:** 1The Lundquist Institute for Biomedical Innovation, Torrance, CA 90502, USAtchuang@lundquist.org (T.-D.C.); 2Department of Obstetrics and Gynecology, David Geffen School of Medicine at University of California, Los Angeles, CA 90095, USA

**Keywords:** non-coding RNA, uterine fibroids, MED12

## Abstract

Uterine fibroids are benign smooth muscle tumors that affect ~70% of women, with Black women being affected at a disproportionate rate. The growth of these tumors is driven by estrogen and progesterone. Driver mutations in genes such as MED12, HMGA2, and FH also play roles in the development and growth of fibroids. Despite their high prevalence, the pathogenesis of fibroids remains largely unknown, leading to a lack of effective therapeutic options. Non-coding RNAs (ncRNAs), including miRNAs (e.g., *miR-21*, *miR-29*, *miR-200*), lncRNAs (e.g., *H19*, *MIAT*, *XIST*), and circRNAs, are important regulatory RNAs that are becoming increasingly implicated in the aberrant expression of protein-coding genes functionally associated with ECM production, cell proliferation, apoptosis, and inflammation in fibroids. Race/ethnicity, MED12 mutations, and ovarian steroids influence the expression of ncRNA expression, further implicating their relevance to fibroid pathogenesis. Therapeutic targeting of these dysregulated ncRNAs in fibroids could enable more precise and individualized non-hormonal-based treatment for this common gynecologic tumor.

## 1. Introduction

Uterine fibroids, or leiomyomas, are benign tumors that develop in the smooth muscle tissue of the uterus. They are among the most common gynecological conditions, affecting approximately 70% of women, with a 3–4-fold higher incidence in Black women as compared with White women. These tumors can cause symptoms such as abnormal uterine bleeding, anemia, pelvic pain, and, in some cases, infertility [1,2,3,4,5,6]. The growth of these tumors is dependent on estrogen and progesterone [7], and currently available medical treatment options are aimed at reducing the levels of estrogen and progesterone, which can cause significant side effects [3,4,8]. These tumors are characterized by aberrant expression of genes regulating the cell cycle, inflammation, ECM composition, and epigenetic regulators [5,9,10]. The mechanisms underlying the aberrant expression of the protein-coding genes regulating these pathways are under intense investigation. Emerging evidence suggests that driver mutations, such as mutations in the genes *MED12* [11,12,13,14], *HMGA2* [15,16,17], or *FH* [18,19,20] play critical roles in fibroid development and growth. Of these driver mutations, *MED12* mutations are the most common and are present in approximately 70% of fibroids [1,11,12,21]. When compared to wild-type fibroids, MED12-mutant fibroids typically develop as multiple tumors rather than a solitary one [1,11] and have a distinct gene profile, showing greater upregulation of genes associated with ECM remodeling and cell cycle progression [12,21]. Fibroids are associated with activation of inflammatory pathways [5,9,10], with upregulation of pro-inflammatory cytokines (TNF-a, IL-6, IFN-γ, etc.) [6,22,23]. This results in widespread DNA damage and epigenetic alterations, driving further tumorigenesis [5,9,10]. Despite the high prevalence of fibroids and their significant impact on women’s health, their pathogenesis remains incompletely understood [1,5,6]. We conducted a comprehensive literature search in PubMed, Scopus, and Google Scholar to identify studies on non-coding RNAs in uterine fibroids. Search terms included combinations of keywords such as long non-coding RNA, lncRNA, microRNA, miRNA, fibroid, leiomyoma, gene expression profiling, and RNA sequencing. Relevant studies were further identified by screening the reference lists of retrieved articles.

## 2. Non-Coding RNAs (ncRNAs)

Recent advances in molecular biology have shed light on the crucial roles of non-coding RNAs (ncRNAs) in gene regulation, epigenetic modification, and tumor pathogenesis [24,25,26]. Historically, fibroid research focused on protein-coding genes, but increasing evidence now underscores the pivotal role ncRNAs have in regulating tumorigenesis [27,28]. In other tumors, ncRNAs act as epigenetic regulators of gene expression, influencing pathways involved in apoptosis, ECM remodeling, and tumor proliferation and infiltration [24,25,29,30,31]. Race and ethnicity can affect the expression of ncRNAs in fibroids differentially [32,33,34,35,36,37] which could account for the racial disparity in symptoms associated with these tumors. ncRNAs are classified into several categories, based on their size and function, as follows:Long non-coding RNAs (lncRNAs): These molecules are greater than 200 nucleotides in length and share many structural features with messenger RNAs (mRNAs), including a 5′ cap and poly(A) tail [38,39,40]. LncRNAs can be classified as ‘intergenic’, ‘intronic’, ‘sense’, or ‘antisense’, depending upon which region of the genome they are transcribed from [38,39,40], and are involved in various regulatory functions, such as chromatin remodeling, gene silencing, and cellular differentiation [38,39,40].
○Pseudogenes: A type of lncRNA that was previously considered as “junk DNA” due to a majority being transcriptionally inactive. However, recent studies have identified some transcriptionally active pseudogenes, suggesting they may play a role in regulating gene expression [41,42,43].○Enhancer RNAs (eRNAs): Enhancer RNAs (eRNAs) are typically considered to be relatively short, ranging from around 50 to 2000 nucleotides in length, with the majority falling within the range of a few hundred nucleotides (around 350 nucleotides) [25,44,45,46]. eRNAs are produced by the bidirectional transcription of enhancer regions in the genome and are identified using the epigenetic modifications H3K27ac, H3K4me1, and H3K27me3 [44,47]; however, there also exists a class of long non-coding eRNAs (elncRNAs) that are unidirectionally transcribed, polyadenylated, and more stable when compared to eRNAs [44].○Super enhancer (SE-lncRNAs): A class of lncRNAs that are transcribed from “super-enhancer” (SE) regions of the genome. When compared to typical enhancers, SEs have a higher density of transcription factors, mediator coactivators, and epigenetic modifications [48,49]. Dysregulation in SE-lncRNAs has also been implicated in oncogenesis, specifically through their regulation of the oncogene MYC [49] and pathways such as TGF-β, MEK/ERK, and Wnt [50,51,52,53].○Circular RNAs (circRNAs): A class of single-stranded, covalently closed ncRNAs that are master regulators of gene expression [54,55,56,57]. Regulation of gene expression is accomplished through a variety of mechanisms, such as sponging miRNAs and serving as a protein scaffold [54,55,56,57]. Recent studies implicate circRNAs in tumor development, making them a possible therapeutic target [54,55,56,57].Small non-coding RNAs (sncRNAs):
○MicroRNAs (miRNAs): Short, approximately 20–22 nucleotides in length, miRNAs regulate gene expression by binding to complementary sequences in target mRNAs, leading to mRNA degradation or translational repression [58,59,60].○Small interfering RNAs (siRNAs): Similar to miRNAs, siRNAs play a role in RNA interference, targeting mRNAs for degradation [25,58].○Piwi-interacting RNAs (piRNAs): piRNAs are involved in the regulation of gene silencing and epigenetic modification through its interactions with the Piwi protein [25,58]. Dysregulation in the expression of many piRNAs has been identified across many tumor types [61].Housekeeping sncRNAs:
○Ribosomal RNA (rRNA), small nuclear RNA (snRNA), small nucleolar RNA (snoRNA), and telomerase RNA are subtypes of sncRNAs involved in basic cellular functions like translation, RNA splicing, and telomere maintenance [25,58].

## 3. ncRNA Mechanisms of Action

### 3.1. miRNAs

miRNA maturation begins in the nucleus, where the primary miRNA transcript (pri-miRNA) is cleaved by the Drosha enzyme to form a precursor miRNA (pre-miRNA) [58,59,60]. After export to the cytoplasm, Dicer processes this pre-miRNA into a mature miRNA duplex, one strand of which is then loaded into the RNA-induced silencing complex (RISC) [58,59,60]. Within RISC, miRNAs typically bind to partially complementary sequences in the 3′ untranslated regions (3′ UTRs) of target mRNAs, resulting in translational repression or mRNA degradation [58,59,60]. Beyond this canonical mechanism, miRNAs can also engage in direct miRNA:miRNA interactions, forming transient duplexes that influence their stability, or in indirect miRNA:miRNA interactions, where multiple miRNAs compete for overlapping binding sites on the same mRNA [58,59,60].

Dysregulated miRNA expression plays a critical role in fibroid development, as these small RNAs regulate pathways involved in cell proliferation, apoptosis, and extracellular matrix (ECM) remodeling [58,59,60,62,63]. When protective miRNAs are lost or repressed, fibrotic and pro-growth genes may be overexpressed, driving excess ECM production, a hallmark of fibroids [58,59,60,62,63]. Conversely, overabundant miRNAs can silence important tumor suppressor genes, further exacerbating fibroid growth and progression [58,59,60,62,63]. For example, in colon cancer, the oncogenic miR-21 was shown to promote tumorigenesis through its indirect regulation of the tumor suppressor miR-145 [64]. The induction of miR-21 was found to induce K-Ras signaling, activating the transcription factor RREBP, which inhibits miR-145 and promotes cancer cell proliferation [64]. Additionally, the expression of several miRNAs, such as let-7a, miR-29a, miR-200c, and miR-222/221 in breast cancer [65,66] and miR-29c [67,68] in fibroids, is influenced by ovarian steroid estrogen and progesterone [65,66,67,68].

### 3.2. lncRNAs

Long non-coding RNAs (lncRNAs) exhibit remarkable regulatory diversity by folding into secondary and tertiary structures that function as molecular scaffolds for chromatin remodelers, transcription factors, and various RNA- or protein-binding partners [29,38,39,69]. Through these scaffold-based interactions, lncRNAs can guide epigenetic modifiers, such as histone acetyltransferases and methyltransferases, to specific genomic regions influencing expression at a transcriptional level [29,38,39,69]. These localized epigenetic modifications may either activate or silence genes, giving lncRNAs a pivotal role in the global regulatory architecture of a cell.

At the post-transcriptional level, lncRNAs can affect mRNA splicing, stability, and translation by binding to RNA-binding proteins and modulating their activity [29,38,39,69]. They also may serve as “miRNA sponges”, competing with endogenous mRNAs for miRNA binding sites, thus reducing miRNA-mediated repression of specific target genes [29,38,39,69]. LncRNAs are also known to promote the creation of enhancer–promoter loops (R-loops), nucleic acid structures each consisting of a DNA:RNA hybrid and a DNA template strand that helps regulate gene transcription and epigenetic modification by acting as a molecular “scaffold” and providing R-loops with stability [29,38,39,69]. LncRNAs not only act as a scaffold, but can also act as the RNA component of R-loops, further driving loop formation [29,38,39,69]. A recent study reported that MED12-mutant fibroids had higher expression of markers for R-loops and R-loop-induced replication stress when compared to MED12-wild-type fibroids [70]. Additionally, dysregulation in R-loop formation due to lncRNA expression, such as the lncRNAs METTL3 [71], HOTTIP [72], and TUG1 [73], has been observed in other tumors.

## 4. ncRNAs in Uterine Fibroids

### 4.1. miRNAs

A growing number of studies have implicated miRNAs in tumor development [58,59,60]. Wang et al. profiled miRNA expression in fibroids by microarray-based analysis [35] and identified 206 well-known miRNAs, 45 of which were found to be differentially expressed when compared to Myo, with the top five most dysregulated miRNAs being let-7, miR-21, miR-23b, miR-29b, and miR-197. The expression of these miRNAs was confirmed by qRT-PCR [35]. In a similar microarray-based profiling study, Zavadil et al. reported that a total of 1117 miRNAs were upregulated and 1557 were downregulated in fibroids, as compared with matched Myo [74]. Of these, the five most upregulated (let-7s, miR-21, miR-23b, miR-27a, and miR-30a) and five most downregulated (miR-29b, miR-32, miR-144, miR-197, and miR-212) miRNAs and their predicted protein targets were identified [74], and they were predicted to interact with fibroid-associated pathways, such as TGF-β [75,76,77], MAPK [78,79,80], Wnt [14,77,81], and JAK/STAT [76,82]. Microarray analysis on 19 fibroids was also performed by Kim et al., confirming the upregulation of let-7c-5p, while additionally confirming the upregulation of miR-181a-5p, miR-127-3p, miR-28-3p, and miR-30b-5p in fibroids [83]. Many other microarray studies have been performed, identifying differentially expressed miRNAs in fibroids [84,85]. A major limitation of microarray analyses is that they do not cover the entire genome and, as such, may not be able to identify key DE-miRNAs [86,87,88]; additionally, the sample size for many of these profiling studies was often small, which may affect the accuracy of the results [84,85,89,90]. Work by Georgieva et al. addressed this issue in fibroids by constructing a comprehensive microRNA profile in two paired fibroid and Myo samples using next-generation sequencing (NGS) [91]. They were able to identify 50 miRNAs that were differentially expressed between fibroid and Myo; however, these results were not confirmed through qPCR [91]. Work by Chuang et al. expanded upon this study and used NGS to create a complete transcriptome profile for all sncRNAs in three paired fibroid and Myo tissue samples [89,90]. They identified 148 differentially expressed (DE) miRNAs [89], with the expression of miR-29c, miR-93, and miR-200c being confirmed through qRT-PCR [92,93,94]. Additionally, Chuang et al. used NGS to identify 15 snoRNAs, 24 piRNAs, 7 tRNAs, and 6 rRNAs that were differentially expressed in paired fibroid and Myo (n = 3) [90]. Of these ncRNAs, they were able to confirm upregulation of five snoRNAs (SNORD30, SNORD27, SNORA16A, SNORD46, and SNORD56), and downregulation of four piRNAs (piR-1311, piR-16677, piR-20365, and piR-4153), one tRNA (TRG-GCC5-1), and one rRNA (RNA5SP202), by qRT-PCR in paired fibroid and Myo (n = 20) [90]. A recent study by the same group validated the differential expression of ten sncRNAs, including five miRNAs (miR-19a-3p, miR-99a-5p, miR-3196, miR-499a-5p, and miR-30d-3p) and five piRNAs (piR-009295, piR-020326, piR-020365, piR-006426, and piR-020485), in fibroid compared to matched Myo (n = 51) using qRT-PCR [95]. All ten sncRNAs were significantly downregulated in fibroid relative to Myo. Among them, miR-19a-3p, miR-3196, miR-30d-3p, piR-006426, and piR-020485 were correlated with MED12 mutation status, while miR-499a-5p and miR-30d-3p were associated with race/ethnicity. These findings highlight the influence of race/ethnicity and MED12 mutation status as key variables affecting sncRNA expression. A list of key dysregulated miRNAs associated with fibroids, and their functions, is presented in Table 1 and Figure 1.

#### 4.1.1. miR-21a-5p

miR-21a-5p is overexpressed in fibroid tissues, and its dysregulation plays a significant role in fibroid pathophysiology [96,97]. One of the key mechanisms by which miR-21a-5p influences fibroid growth is through its actions on apoptotic pathways. In fibroids, miR-21a-5p has been found to promote fibroid cell survival by concurrently suppressing apoptosis via inhibition of PDCD-4 (Programmed Cell Death 4) [96], a crucial apoptosis regulator, and enhancing ECM remodeling through upregulation of TGF-β3 and MMP-2/11 [97,98].

#### 4.1.2. miR-29 Family

The miR-29 family, comprising miR-29a, miR-29b, and miR-29c, plays a crucial role in fibroid pathogenesis, primarily through influences on extracellular matrix (ECM) remodeling and fibrotic pathways [67,92,99,100,101]. The miR-29 family is consistently downregulated in fibroid tissues, and its suppression has been linked to the excessive ECM deposition characteristic of fibroids [67,92,99,100,101]. A luciferase assay indicated that TGF-β3 was a direct target of miR-29c [92], and in vitro treatment of fibroid cells with a miR-29c agonist resulted in significant decreases in *TGF-β3* mRNA and in the protein targets of miR-29c including COL1A1 and COL3A1 [92]. Additionally, when miR-29c was inhibited by miR-29c siRNA, there was increased fibroid cell proliferation and upregulation of key ECM remodeling enzymes, including MMP-2 and MMP-9 [100]. The inhibition of miR-29c also promoted expression of STAT3 and the cell cycle regulators [102,103,104] cyclin D1 and c-Myc [101]. Additionally, miR-29c interacted with DNMT3A [67], which is critical for maintaining DNA methylation patterns and gene silencing [105,106]. Inhibition of both DNMT3A and p65 was shown to restore miR-29c expression in fibroid cells, thereby mitigating the fibrotic effects of TGF-β3 [67]. Treatment of fibroid cells in vitro with Tranilast, an anti-inflammatory drug, induced miR-29c expression, along with downregulation of its targets, including TGF-β3, the cell cycle regulators [102,104] CCND1 and CDK2, the ECM components [107,108] COL1 and COL3A1, and epigenetic regulators [106,109] DNMT1 and EZH2 [110]. Furthermore, TGF-β3 was shown to upregulate DNMT1, resulting in methylation of the miR-29c promoter and its downregulation, which, in turn, led to upregulation of TGF-β3, thus creating a positive feed forward fibrotic feedback loop [92].

*miR-29b*, another member of this family, is downregulated in fibroids and has been shown to be activated in a reactive oxygen species (ROS)-dependent manner [68,100], linking oxidative stress to cellular senescence. This activation potentially involves the AKT signaling pathway, as AKT is a known mediator of ROS-associated responses [68]. Cellular senescence induced by miR-29b may contribute to a growth-limiting effect in fibroids, adding another layer of complexity to its role in fibroid biology [68]. Additionally, an in vivo study performed by Qiang et al. found that, when *miR-29b* was restored in fibroid xenografts, it resulted in the inhibition of collagen accumulation and solid tumor formation [68]. Qiang et al. further reported that xenografts generated from Myo cells in vivo, transduced with a *miR-29b* knockdown lentiviral vector, resulted in increased expression of COL1A1, COL1A2, COL3A1, COL5A1, COL5A3, and COL7A1 [68]. Additional evidence demonstrating the key role of the miR-29 family in fibroid pathogenesis is the demonstration of its hormone responsiveness, with estrogen and progesterone inhibiting the expression of *miR-29b/c* in fibroids both in vivo [68] and in vitro [67], providing another potential mechanism for the reduced expression of this important miRNA in fibroids.

#### 4.1.3. miR-200c

miR-200c, a member of the miR-200 family, is significantly downregulated in fibroids [33,36,83,94,111] in a race-dependent manner, with lower expression in tumors from Black women as compared with White women [36]. Its reduced expression has been linked to the promotion of epithelial-to-mesenchymal transition (EMT)-related genes [36,83,94,111]; specifically, *miR-200c* upregulates E-cadherin, a crucial protein in maintaining epithelial cell integrity through repression of transcription factors ZEB1/2 [36]. When *miR-200c* levels were restored in fibroid cells, E-cadherin expression was upregulated, resulting in the reversal of EMT-related phenotypic changes and decreased fibroid cell proliferation [36]. Another in vitro study found that, when *miR-200c* was overexpressed, it induced cellular senescence (measured by % of β-Galactosidase positive cells) [83]. The same study also reported that, when *miR-181a* and *miR-182* were overexpressed [83], cellular senescence was induced and the expression of AKT3 and CCND2 was reduced [83].

In addition to its role in regulating EMT, *miR-200c* also modulates inflammatory pathways, which are critical for fibroid progression. Studies in other tumors, such as breast and ovarian cancers, have demonstrated that *miR-200c* is an important promoter of the NF-κB signaling pathway [112,113,114], which is a central mediator of inflammation and cell survival that, when induced, promotes tumorigenesis [112,113,114]. Chuang et al. showed, in an in vitro study, that the induction of miR-200c resulted in decreased phosphorylation of IkBα [94], an NF-κB inhibitor [114], and decreased nuclear translocation of RelA/p65 [94], a protein subunit of NF-κB [114]. The reduction in p65 translocation also decreased the transcriptional activity at the *IL8* promoter and increased caspase-3/7 activity [94]. Furthermore, treatment of fibroid cells with siRNA to RelA/p65 induced miR-200c expression [111]. In addition, Tranilast, an anti-inflammatory drug with beneficial therapeutic effects in fibroids [115,116], induced *miR-200c* expression in fibroids and Myo, with a stronger effect in fibroids when compared to Myo [111]. Although treatment of fibroid cells with Tranilast did not reduce RelA/p65 levels, the phosphorylation of RelA/p65 was reduced, thereby reducing its ability to bind to the miR-200c promoter and resulting in the induction of miR-200c [111]. Expression of IL-8, a pro-inflammatory cytokine associated with increased cell migration and immune response [117,118], and CDK2, a cell cycle regulator [102,104], were also reduced by treatment of fibroid cells with Tranilast [117,118]. CDK2, along with CCND2 and E2F1, were confirmed to be targets of miR-200c by luciferase assay [33]. Overexpression of miR-200c resulted in the inhibition of CDK2 mRNA and protein expression, while inhibition of miR-200c had the opposite effect [33].

#### 4.1.4. miR-93

*miR-93* is another miRNA that is downregulated in fibroids and its expression was inversely correlated to the expression of key cell cycle regulators [33,102,104], such as CDK2, CCND1, and E2F1 [33,93]. Furthermore, when miR-93 was induced through transfection of agomir, the expression for CCND1 and E2F1 mRNA/protein was reduced [33]. Lentiviral induction of *miR-93* in vitro directly repressed F3 and IL8 expression and indirectly repressed CTGF and PAI-1 through its effects on F3 [93]. Moreover, when *miR-93* was upregulated, it repressed the expression of F3 and IL8, which are key regulators of inflammation in fibroids [93,117,118,119]. Functional studies have also shown that increased miR-93 expression results in increased caspase-3/7 activity, signaling the activation of apoptosis pathways [93].

#### 4.1.5. hsa-let-7 Family

The *let-7* family of miRNAs, which includes several members, such as *let-7a/b/c*, is severely dysregulated in fibroids [17,35,120,121,122], generally being overexpressed in small fibroids and underexpressed in large fibroids [121], with the exception of *let-7c*, which was consistently upregulated in fibroids [121], and let-7a, which was consistently downregulated in fibroids [17]. Dysregulation in *let-7* family expression is believed to contribute to fibroid growth through its effects on HMGA2 expression [17,35,121,122], a protein that promotes fibroid cell proliferation and is overexpressed in ~20% of fibroids [16,123]. Luciferase assay indicated that HMGA2 is a target for *let-7* [121]. The overexpression of the let-7 family was associated with the repression of HMGA2 and increased fibroid cell proliferation [121]. In contrast, downregulation of the let-7 family was associated with upregulation of HMGA2 [121]. Fibroids with truncated binding sites of HMGA2 for let-7 had increased HMGA2 expression [122], further implicating the role of let-7 in HMGA2 regulation. *Let-7c/d/e/f-2* expression levels were also associated with higher levels of senescence, as measured by % of SA-β-Gal positive cells, and reduced cell proliferation, as measured by % of Ki-67-positive staining cells [120]. Another report linked *miR-542-3p* expression in fibroids with cell proliferation and apoptosis [124]. Overexpression of *miR-542-3p* in vitro inhibited cell proliferation through induction of G1 and G2/M cell cycle arrest. Furthermore, survivin, an apoptotic inhibitor [125,126], was shown to be a target of *miR-542-3p* by luciferase assay [124].

#### 4.1.6. miR-197

miR-197 is consistently downregulated in fibroids [127,128,129], which was confirmed through qRT-PCR, and its downregulation is associated with increased cell proliferation and the induction of cell cycle arrest [127,129]. When *miR-197* was induced in fibroid cells, it was shown to directly target IGFBP5, inhibiting its expression and cell proliferation [127]. IGFBP5 was also confirmed to be a target of *miR-197* by luciferase assay.

#### 4.1.7. miR-199a-5p

*miR-199a-5p*, a member of the *miR-199* family, is downregulated in fibroids, more so in fibroids possessing MED12 mutations [130]. This miRNA plays a significant role in regulating cell proliferation, apoptosis, and fibrosis in other tumors [131,132]. Restoring the expression of *miR-199a-5p* in fibroid cells has been shown to suppress cell proliferation and induce apoptosis, which suggests its potential as a therapeutic target [130]. Additionally, *miR-199a-5p* regulates multiple signaling pathways involved in fibrosis, including the TGF-β pathway [130].

#### 4.1.8. miR-139-5p

*miR-139-5p* is significantly downregulated in fibroids, and its reduced expression has been linked to increased ECM contractility and enhanced fibroid cell migration [133]. In fibroids, miR-139-5p has been implicated in regulating COL1A1, a major ECM component [133]. Overexpression of *miR-139-5p* resulted in downregulation of COL1A1 expression and a reduction in phosphorylated p38 MAPK [133].

#### 4.1.9. miR-150-5p

*miR-150-5p* was shown to be downregulated in fibroids through qPCR [134]. When cells were transfected with an *miR-150-5p* mimic in vitro, the expression of Akt protein, which modulates p27Kip1, was reduced [134].

### 4.2. lncRNAs

Like miRNAs, lncRNAs have also emerged as key regulators of gene and epigenetic regulation [38,39,40]. A growing number of studies have highlighted the role of lncRNAs in tumorigenesis [31,135,136,137]. The lncRNA expression profile for fibroid and Myo tissues (n = 35) was first reported by Guo et al., which identified 816 DE-lncRNAs (527 up, 289 down) using a microarray [138]. The DE-lncRNAs identified were positively correlated with tumor size, and their corresponding cis mRNA expression [138]. In this study, the upregulation of AK023096 and downregulation of *CAR10* were confirmed by qPCR [138]. Chuang et al. was the first group to use NGS to identify DE-lncRNAs in paired fibroid and Myo tissues (n = 3), revealing 5941 DE-lncRNAs (2813 up, 3128 down) [89]. qRT-PCR confirmed upregulation of *lnc-MEG3* and *LINC00890* and downregulation of *HULC*, *TSIX*, *lnc-KLF9-1*, and *lnc-POTEM-3* [89]. In a more recent study, Chuang et al. used NGS in a larger sample set (n = 19) and validated several lncRNAs by qPCR in a large number of fibroids and matched Myo (n = 69) [32]. They confirmed the differential expression of 15 lncRNAs, when comparing fibroids to matched Myo, with expression of *TPTEP1*, *PART1*, *RPS10P7*, *MSC-AS1*, *SNHG12*, *CA3-AS1*, *LINC00337*, *LINC00536*, *LINC01436*, *LINC01449*, *LINC02433*, and *LINC02624* being higher in fibroids, and expression of *ZEB2-AS1*, *LINC00957*, and *LINC01186* being lower in fibroids [32]. Meng et al. also used RNAseq to identify an additional 553 DElncRNAs (283 up, 270 down) when comparing matched fibroid and Myo tissue samples (n = 10) [139]. These findings were not confirmed through qPCR, used a small sample size (n = 10), and did not record for *MED12* mutation status, which may affect the accuracy of the results.

Chuang et al. further examined the effects that race/ethnicity and *MED12* mutation status had on lncRNA expression in fibroids. NGS showed that 63 lncRNAs were *MED12*-dependent (>1.5-fold change in MED12-mut but not wild-type) and 65 lncRNAs were race-dependent (>1.5-fold change in Black but not White) [32]. They then used qRT-PCR to confirm the expression of 5 upregulated race-dependent lncRNAs (*RPS10P7*, *SNHG12*, *LINC01449*, *LINC02433*, and *LINC02624*), 10 upregulated MED12-dependent lncRNAs (*TPTEP1*, *PART1*, *RPS10P7*, *MSC-AS1*, *LINC00337*, *LINC00536*, *LINC01436*, *LINC01449*, *LINC02433*, and *LINC02624*), and 1 MED12-dependent downregulated *LINC01186* [32]. In a later investigation, the same research group compared lncRNA expression profiles in fibroids from premenopausal and postmenopausal women using NGS. This analysis uncovered 62 lncRNAs that were specifically dysregulated in the postmenopausal cohort [140]. To validate these findings, nine lncRNAs were selected for validation by PCR in an expanded cohort of 31 postmenopausal and 84 premenopausal paired samples. The results showed that LINC02433, LINC01449, SNHG12, H19, and HOTTIP were elevated in premenopausal fibroids, but remained unchanged in postmenopausal ones, while ZEB2-AS1 displayed higher levels only in postmenopausal fibroids. CASC15 and MIAT were consistently upregulated in both groups, although the magnitude of increase was lower in the postmenopausal group. In contrast, LINC01117 was markedly reduced in postmenopausal fibroids, with no significant alteration in premenopausal tissues [140]. Further analysis based on MED12 mutation status revealed that lncRNAs such as LINC01449, CASC15, and MIAT exhibited weaker or diminished differential expression between mutation-positive and mutation-negative samples in postmenopausal compared to premenopausal groups [140]. A pie chart depicting a more detailed distribution of lncRNAs based on Chuang’s study [32] in fibroids is shown in Figure 1. In another study Akbari et al. reported that the expression of lncRNA *SRA1* is MED12-dependent, with MED12 mutants having higher *SRA1* expression [141]. There are a scant number of studies that have addressed the functional roles of differentially expressed lncRNAs in fibroids, as outlined below and summarized in Table 2.

#### 4.2.1. H19

*H19* is a long non-coding RNA (lncRNA) that is overexpressed in fibroids, particularly in those that exhibit altered expression of MED12 and HMGA2 [142,143,144,145]. In an in vitro experiment in fibroids, knockdown of *H19* inhibited TGF-β signaling through downregulation of TGFBR2 and TSP1, while also upregulating key ECM components, including COL3A1, COL4A1, and COL5A2 [142]. These effects of *H19* were blocked by knockdown of TET3, suggesting that *H19* acts through TET3 to regulate gene expression [142]. H19 is known to promote the expression of genes involved in fibroid growth, such as MED12, HMGA2, and other extracellular matrix (ECM) genes [142,143,145]. Both *H19* and TET1, a key regulator of DNA methylation [106,146], were identified to be strong predictive markers for postoperative recurrence of fibroids [143], suggesting that levels of *H19* and TET1 could potentially be used as diagnostic/predictive markers [143]. One of the key features of H19 is the identification of single nucleotide polymorphisms (SNPs) in its sequence, which may contribute to the development of larger fibroids and increase the risk of the malignant leiomyosarcoma [142,145].

A SNP associated with lncRNA *TCONS_l2_00000923*, which is upregulated in fibroids in a race-dependent manner [37], with Black women having higher expression compared to White women, is also associated with fibroid cell proliferation and mutations in FH [37]. Another study found that *HOTAIR*, another lncRNA, had many gene polymorphisms that correlate with fibroid susceptibility [147] (*rs920778* was associated with reduced risk and *rs12826786* associated with increased risk). In this study, the frequency of the CTGA haplotype of *HOTAIR* was lower, while the frequency of the CCGA, TCGA, TTTA, and TTGA haplotypes was higher in fibroids [147].

#### 4.2.2. MIAT

*MIAT* (myocardial infarction-associated transcript) is another lncRNA that is overexpressed in fibroids, with significantly higher levels in MED12-mutated fibroids [145,148,149]. This overexpression was independent of race/ethnicity, suggesting it is a consistent marker of fibroid pathology across different populations [148]. *MIAT* functions primarily as a miRNA sponge for the *miR-29* family [149]. When *MIAT* was knocked down by siRNA in fibroid cells, the levels of *miR-29* family miRNAs increased, resulting in downregulation of protein-coding genes targeted by this miRNA, including COL1A1, COL3A1, and TGF-β3 [149]. These genes are crucial for ECM production and fibroid growth, making *MIAT* an important lncRNA in fibroid pathogenesis [149]. Furthermore, in an in vivo study using human-derived fibroid xenografts implanted in SCID mice, *MIAT* knockdown via lentiviral delivery resulted in a 30% reduction in tumor weight [148], upregulation of *miR-29* family expression, and downregulation of *miR-29*’s targets, including TGF-β3, FN1 (fibronectin), and COL3A1 [148], in the xenografts. Additionally, the expression of cell cycle regulatory genes, such as CCND1, CDK2, and E2F1, was significantly reduced, further supporting the potential of MIAT as a therapeutic target for fibroid treatment [148].

#### 4.2.3. XIST

*XIST* was reported by some studies to be significantly upregulated in fibroids compared to matched myometrial tissues [150,151,152], acting as a miRNA sponge for *miR-29c* and *miR-200c* [150]. *XIST* expression could also be induced by estradiol, progesterone, and their combination [150]. When *XIST* was inhibited in vivo by siRNA, there was a 15% reduction in tumor weight and increased expression of *miR-29c* and *miR-200c* [150,151]. The targets of both *miR-29c/200c* were impacted following inhibition of *XIST*, with a significant decrease in *miR-29c*’s targets COL3A1, TGF-β3, CDK2, and SPARC, and *miR-200c*’s targets, including CDK2, FN1, and TDO2 [150,151]. In contrast to these findings, Sato et al. reported that *XIST*, measured by qPCR, was downregulated in fibroids (n = 11) and was associated with hypomethylation on the X chromosome [152]. A potential explanation for the discrepancy between the expression profiles outlined by Chuang et al. [150,151] and those by Sato et al. [152] was attributed to the use of different internal controls used for qPCR between the two studies [150,151] (FBXW2 was used by Chuang [150,151]; GAPDH was used by Sato [152]).

#### 4.2.4. LINCMD1

*LINCMD1* is significantly downregulated in fibroids and is known to act as a sponge for miR-135b, an important miRNA involved in ECM regulation [153]. Luciferase assays confirmed that *LINCMD1* directly interacts with miR-135b [153]. When *LINCMD1* was knocked down, there was a significant increase in miR-135b levels, and a corresponding reduction in APC expression, leading to β-catenin accumulation and increased COL1A1 expression [153]. These effects were independent of race/ethnicity [153].

#### 4.2.5. lnc-AL445665.1-4

*lnc-AL445665.1-4* is upregulated in fibroid tissues, particularly in patients with multiple uterine leiomyomas [154]. This lncRNA was shown by luciferase assay to bind to *miR-146b-5p* [154]. Silencing of *lnc-AL445665.1-4* inhibited fibroid cell proliferation and induced expression of *miR-146b-5p* [154], further indicating its role in regulating fibroid cell growth and survival.

**Table 1 cells-14-01290-t001:** List of Dysregulated miRNAs in Fibroids with Identified Function.

miRNA	Function	Expression	Location	Reference(s)
miR-181a-5p	* Positively associated with cellular proliferation, ECM turnover, angiogenesis, and TGFBR2/IGF2BP1 expression * Upregulation induces cellular senescence and represses AKT3 in spheroid cultures.	Upregulated	chr1:198,859,044-198,859,153	[83]
miR-127-3p miR-28-3p miR-30b-5p	* Positively associated with cellular proliferation, ECM turnover, angiogenesis, and TGFBR2/IGF2BP1 expression	Upregulated	chr14:100,882,979-100,883,075 chr3:188,688,781-188,688,866 chr8:134,800,520-134,800,607	[83]
let-7c	* Positively associated with cellular proliferation, ECM turnover, angiogenesis, and TGFBR2/IGF2BP1 expression * Luciferase assay shows HMGA2 as a target * Overexpression associated with reduced expression of HMGA2 * Fibroids with higher proportions of truncated binding sites for let7 in HMGA2 had higher HMGA2 expression * Inversely correlated with Ki67	Upregulated	chr21:16,539,828-16,539,911	[17,35,120,121,122]
let-7a	* Luciferase assay shows HMGA2 as a target * Overexpression associated with reduced expression of HMGA2 * Fibroids with higher proportions of truncated binding sites for let7 in HMGA2 had higher HMGA2 expression	Downregulated	chr9:94,175,957-94,176,036	[17,35,120,121,122]
miR-29a	* Overexpression results in downregulation of fibrillar collagens * Inhibition promotes expression of ECM remodeling genes	Downregulated	chr7:130,876,747-130,876,810	[67,92,99,100,101]
miR-29b	* Overexpression nduces cellular senescence in spheroid cultures. * Overexpression results in downregulation of fibrillar collagens (COL1A1, COL2A1, COL3A1) * Inhibition promotes expression of ECM remodeling genes	Downregulated	chr7:130,877,459-130,877,539	[67,92,99,100,101]
miR-29c	* Targets cell cycle regulatory protein CDK2. * Inverse relationship with TGF-B3. * Luciferase assay confirms TGF-B3 as a target * Overexpression results in downregulation of fibrillar collagens (COL1A1, COL2A1, COL3A1) * Inhibition promotes expression of ECM remodeling genes * Overexpression inhibits protein/mRNA expression of COL3A1 and DNMT3A, secreted COL3A1, and rate of cell proliferation. * Knockdown of p65 induced expression. * Treatment with Tranilast decreased expression	Downregulated	chr1:207,801,852-207,801,939	[67,92,99,100,101]
miR-200c	* Race-dependent, with lower expression in fibroids of Black patients when compared to White * Induced cellular senescence in spheroid cultures when overexpressed * Luciferase assay shows CDK2, CCND1, and E2F1 as targets * Overexpression downregulated mRNA and protein expression of CDK2 * Upregulation repressed ZEB1/2, Vimentin, and fibroid cell proliferation and increased E-cadherin. * Microarray assay confirmed TIMP2, FBLN5, and VEGFA as targets for miR-200c. * Overexpression resulted in decreased IKBKB phosphorylation and p65 transcriptional activity at the IL8 promoter while increasing caspase-3/7 activity.	Downregulated	chr12:6,963,699-6,963,766	[33,36,94,111,112,113]
miR-93	* Luciferase assay shows CDK2, CCND1, and E2F1 as targets * Overexpression downregulated mRNA and protein expression of *CCND1* and *E2F1* * Fibroids express significantly higher levels of its host gene MCM7. * Upregulation F3, CTGF, PAI-1, and IL8 expression.	Downregulated	hr7:100,093,768-100,093,847	[93]
miR-21	* Increased expression results in upregulation of fibronectin, COL1A1, CTGF, Versican and DPT. * Knockdown results in increased expression of apoptotic markers PDCD-4 and Caspase-3. * Knockdown results in increased expression of EEF2, a marker of cell proliferation * Induction through lentiviral infection induced expression of TGF-β and MMP-2/11	Upregulated	chr17:59,841,266-59,841,337	[96,97]
miR-199a-5p	* Regulate cell proliferation and apoptosis in-vitro. * Bioinformatics showed MED12 as a potential target for miR-199a-5p. * MED12 dependent, with MED12 mutants having lower expression	Downregulated	chr19:10,817,426-10,817,496	[130]
miR-139-5p	* Restored expression results in decreased ECM contractility, cell migration * Restored expression reduced protein expression of COL1A1 and phosphorylated p38 MAPK.	Downregulated	chr11:72,615,063-72,615,130	[133]
miR-542-3p	* Luciferase assay shows survivin as a predicted target. * Overexpression inhibits cell proliferation through induction of G1 and G2/M cell cycle arrest.	Downregulated	chrX:134,541,341-134,541,437	[124]
miR-150-5p	* After transfection of cultured cells with miR-150 mimic, expression levels of its predicted target AKT decreased while p27Kip1 levels increased.	Downregulated	chr19:49,500,785-49,500,868	[134]
miR-135b	* Confirmed to be target for LINCMD1 by Luciferase Assay * After LINCMD1 knockdown, expression was induced	Downregulated	chr1:205,448,302-205,448,398	[153]
miR-146b-5p	* Targeted by Lnc-AL445665.1–4 by Luciferase Assay * Silencing Lnc-AL445665.1–4 negatively regulated miR-146b	Downregulated in MUL Upregulated in SUL	chr10:102,436,512-102,436,584	[154]
miR-197	* Decreased expression associated with increased fibroid cell proliferation and induction of cell cycle arrest * Luciferase assay shows IGFBP5 as a target	Downregulated	chr1:109,598,893-109,598,967	[35,74,127,128,129]

#### 4.2.6. CAR10

*CAR10* (Chromatin-associated RNA Intergenic 10), was confirmed through qPCR to be upregulated in fibroids [138]. Knockdown of *CAR10* reduced fibroid cells’ ability to proliferate in vitro [138]. Furthermore, following *CAR10* knockdown, the expression of its neighboring protein-coding gene, ADAM10, was reduced.

#### 4.2.7. APTR

*APTR* (Adenylate Phosphoribosyl Transferase RNA) is another lncRNA that is upregulated in fibroids [155]. Overexpression of *APTR* led to increased fibroid cell proliferation in both in vivo and in vitro models and increased expression of proteins involved in the Wnt/β-catenin pathway, which is implicated in fibroid tumorigenesis [155]. Additionally, ERα (Estrogen Receptor Alpha) is predicted to be a target of *APTR*, and when ERα was knocked down in fibroid cells, it led to overexpression of *APTR* and prevented the upregulation of Wnt pathway proteins [155].

A sequencing study conducted by Chuang et al. (GSE224991) did not find the expression of *lncSRA1* and *APTR* to be differentially expressed in fibroids, which could be related to differences in study populations.

## 5. Super Enhancer lncRNAs (SElncRNAs)

Super enhancers (SEs) are extensive clusters of transcriptional enhancers that exhibit unusually high levels of transcription factor binding and co-activator recruitment, driving the robust expression of genes critical for cellular identity [48,49,156]. In various tumor types, aberrant SE activity can amplify oncogene expression, accelerating tumor growth and progression [48,49,156]. Super enhancer-associated lncRNAs (SE-lncRNAs) are a subclass of long non-coding RNAs transcribed from these SE regions [50,51,52]. Rather than the SEs themselves, it is these SE-lncRNAs that exert significant influence on tumorigenesis by stabilizing or enhancing the transcriptional machinery at oncogenic loci, sequestering regulatory proteins, or modulating nearby gene expression networks [48,49,156]. Through action in either cis (locally) or trans (at a distance), SE-lncRNAs can regulate entire sets of genes, thereby promoting the pro-growth or pro-survival pathways that drive tumor development and maintenance [48,49,156].

In the only study to date examining SE-lncRNA expression in fibroids, Chuang et al. investigated the expression profiles of these transcripts in fibroids and paired Myo (n = 8) [34]. Microarray analysis identified 721 SE-lncRNAs that were upregulated, and 247 that were downregulated, in fibroids when compared to Myo [34]. Of these, a subset demonstrated race-dependent alterations, while another subset was associated with MED12 mutation status, suggesting that both genetic background and mutational profiles can shape SE-lncRNA dysregulation [34]. Moreover, 13 protein-coding genes located adjacent to these SE-lncRNAs displayed similarly altered expression, implying that SE-lncRNAs may directly regulate nearby genes involved in fibroid pathogenesis [34]. Currently, there are no functional studies on any of the differentially expressed SElncRNAs.

## 6. circRNA

CircRNAs are a subtype of lncRNAs that form a covalently closed loop structure, generated through the process of back-splicing [54,55,56]. This structure results in multiple copies of RNA transcripts due to circular replication, which makes circRNAs more stable than their linear counterparts. First discovered in humans in 1991, they were initially considered rare and non-functional, but recent studies have revealed their significant roles in gene regulation, particularly in other tumors [55,56,157,158,159,160]. CircRNAs regulate gene expression through several mechanisms. One of the most studied roles is their ability to act as miRNA sponges, thereby sequestering miRNAs and preventing them from inhibiting their target mRNAs [57]. Additionally, circRNAs can modulate the stability of other RNA species, including long non-coding RNAs (lncRNAs) and messenger RNAs (mRNAs), contributing to the regulation of various cellular processes [57,158,159,160]. Moreover, circRNAs regulate transcription and form R-loops, which are structures that can influence gene expression through interactions with chromatin [158]. They also participate in DNA methylation processes, affecting epigenetic regulation [158].

In a small subset of tissues (n = 5) microarray was used to profile the expression of circRNAs in fibroids (all wild-type) and their matched Myo [160]. This analysis showed 579 upregulated and 625 downregulated circRNAs [160], with upregulation of only *hsa_circRNA_0056686* being confirmed through qRT-PCR [159,160]. *Hsa_circRNA_0056686* was also upregulated in tumor-associated fibroblasts (TAFs) from the fibroid microenvironment and was positively correlated with tumor size [159]. Reducing the levels of *hsa_circ_0056686* by shRNA in TAFs derived from fibroids resulted in inhibition of TAF cell proliferation, suppression of apoptosis, and induction of expression of ECM and endoplasmic reticulum proteins [159]. The same study also reported that *miR-515-5p* targets *hsa_circ_0056686*, confirmed by luciferase assay, and furthermore, when fibroid cells were treated with *hsa_circ_0056686* shRNA in vitro, it resulted in the overexpression of *miR-515-5p* and restored *hsa_circ_0056686* malignant behaviors [159]. This suggests that targeting *hsa_circ_0056686* may provide therapeutic potential for modulating fibroid growth and fibrosis [159].

**Table 2 cells-14-01290-t002:** List of Dysregulated IncRNAs in Fibroids with Identified Function.

lncRNA	Function	Expression	Aliases	Location	Reference(s)
H19	* Promotes expression of MED12, HMGA2, and many key ECM remodeling genes * Inverse relationship with TET1. * Uses TET3-mediated epigenetic mechanism to alter gene expression * Strong predictive marker for preoperative recurrence of fibroids * Upregulation induces TGFBR2 and TSP1 expression	Upregulated	ASM BWS WT2 ASM1 D11S813E MIR675HG LINC00008 NCRNA00008	chr11:1995130-2001710	[142,143]
MIAT	* MED12-dependent, with higher expression in MED12 mutant fibroids when compared to wild-type * Luciferase assays shows miR-29 as a target * Inhibition resulted in downregulation of COL1A1, COL3A1, TGFB3 * Knockdown in fibroid xenografts resulted in reduction of tumor weight, cell proliferation, expression of cell cycle regulatory genes (CCND1, CDK2, E2F1) and increased expression of the miR-29 family * Knockdown reduced mRNA/protein expression of COL3A1, FN1, TGFB3 and total collagen protein.	Upregulated	RNCR2 GOMAFU C22orf35 LINC00066 NCRNA00066 lncRNA-MIAT	chr22:26646428-26851957	[148,149]
XIST	* Expression induced by 17β-Estradiol, progesterone and their combination. * Knockdown in-vitro resulted in decreased cell proliferation and increased expression of miR-29c, miR-200c. The downstream targets of miR-29c and miR-200c were also downregulated. * Immunoprecipitation analysis shows miR-29c and miR-200c as targets * Fibroid xenografts treated with siRNA for XIST in-vivo resulted in a significant reduction of tumor weight and increased expression of miR-29c and miR-200c * Knockdown significantly reduced total collagen protein, COL3A1, FN1, CDK2, SPARC, EZH2, apoptotic marker caspase-3, and the cell proliferation marker Ki67. * Associated with hypomethylation on X chromosome	Upregulated	SXI1 swd66 DXS1089 DXS399E LINC00001 NCRNA00001	chrX:73817775-73852753	[150,151,152]
lnc-AL445665.1-4	* Luciferase assay shows miR-146b-5p as a target * Inhibition reduced fibroid cell proliferation and miR-146b-5p expression.	Downregulated in SUL Upregulated in MUL	lnc-CBWD5-4:7 NONHSAT131696	chr9:65218523-65219575	[154]
APTR	* Overexpression induced tumor cell proliferation and colony formation both in vitro and in vivo. * Functional assays showed ERα as a target of APTR * Overexpression induced expression of Wnt pathway proteins.	Upregulated	RSBN1L-AS1	chr7:77477984-77697345	[155]
LINCMD1	* Luciferase assay shows miR-135b as target * Knockdown increased levels of miR-135b, accumulation of β-catenine, increased expression of COL1A1, and reduced expression of APC.	Downregulated	LINC-MD1 MIR133BHG	chr6:52146814-52151425	[153]
CAR10	* Knockdown inhibited proliferation of fibroid cells in vitro and downregulated its neighboring gene *ADAM12*	Upregulated	ADAM12 MCMP MLTN MLTNA MCMPMltna ADAM12-OT1	chr22:37469068-37472724	[138]
TCONS_l2_00000923	* Upregulation associated with downregulation of PLD5 and increased tumor size * Race-dependent, with Blacks having increased expression when compared to Whites	Upregulated	-	-	[37]
HOTAIR	* Differential effect depending on gene polymorphisms * CTGA haplotype downregulated in fibroids, but CCGA, TCGA, TTTA, and TTGA haplotypes were upregulated	Depends on Haplotype	HOXAS HOXC-AS4 HOXC11-AS1 NCRNA00072	chr12:53962308-53974956	[147]
Hsa_circ_0056686	* Upregulation correlates with fibroid size * Upregulated in tumor associated fibroblasts (TAFs) * TAFs transfected with Hsa_circ_0056686 shRNA were unable to proliferate and induce expression of ECM proteins * Luciferase assay confirms that it is a target of miR-515-5p * miR-515-5p overexpression in TAF media containing Hsa_circ_0056686 shRNA restored Hsa_circ_0056686's maligant behaviors	Upregulated	-	-	[159,160]

## 7. Conclusions

Non-coding RNAs (ncRNAs), specifically long non-coding RNAs (lncRNAs), microRNAs (miRNAs), and circular RNAs (circRNAs), play critical roles in the pathophysiology of uterine fibroids through their roles in regulating the expression of protein-coding genes, which are aberrantly expressed in fibroids. These ncRNAs regulate key processes such as extracellular matrix (ECM) remodeling [92,97,100,143,149], cell proliferation [127,130,155,159], apoptosis [93,96,130], and fibrosis [96,130,159], which are all implicated in fibroid development and progression. The expression of some ncRNAs is influenced by estrogen/progesterone [92,148], race [32,36,92,95] and MED12 mutation status [32,95,141,142,149]. In general, tumors from Black patients and those with MED12 mutations exhibit heightened misexpression of specific ncRNAs, which could lead more aberrant expression of protein-coding genes and an etiology for differential symptom severity [32].

Currently, there are no functional studies on the roles of other ncRNA classes, such as small nucleolar RNAs (snoRNAs), Piwi-interacting RNAs (piRNAs), and tRNA-derived fragments, in fibroid biology, although some have been shown to be differentially expressed in fibroids [90]. Future research into these less well-characterized ncRNA classes could reveal additional layers of regulation of protein-coding genes. Given the significant role of ncRNAs in fibroid biology, targeting these molecules as a therapeutic strategy holds great promise [161,162]. In fact, the lncRNAs *MIAT* and *XIST* have been shown to be potential therapeutic targets in fibroids [145,148,149,150,151,152]. Targeting ncRNAs for therapeutic purposes could be utilized in gene therapy strategies aimed at correcting the dysregulated ncRNAs [148,151], or in drugs [110,111]. Furthermore, because race/ethnicity and MED12 mutation of the tumor influence the expression of ncRNAs [32,36,92,141,142,149], individualized therapies could be developed based on these variables.

## Figures and Tables

**Figure 1 cells-14-01290-f001:**
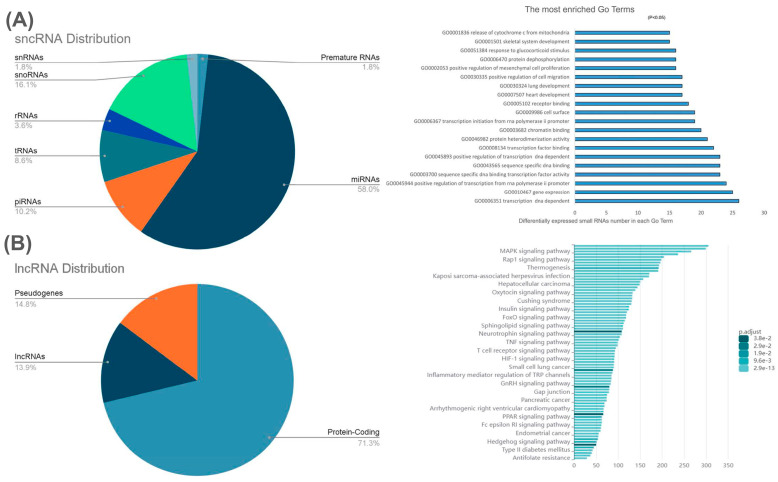
Pie charts depicting the relative percent of different classes of ncRNAs in fibroids based on NGS data (GSE100338) [90] and (GSE224991) [32]. The pie charts represent (**A**) relative proportion of DEsncRNAs (>1.5-fold difference) between fibroids and matched Myo, along with the associated pathways they may regulate to the right and (**B**) the relative proportion of DElncRNAs (>1.5-fold difference) between fibroids and matched Myo, along with the associated pathways they may regulate to the right. Note that in (**B**), the category “Other” refers to DElncRNAs that were not classified as intergenetic, intronic, antisense, or processed lncRNAs.

## Data Availability

Supporting information is available from the corresponding author, O.K., on request.

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
