# Peer review of "The Roles of Non-Coding RNAs in the Pathogenesis of Uterine Fibroids"

_cells, 2025, doi:10.3390/cells14161290_

Round 1
Reviewer 1 Report
Comments and Suggestions for Authors
This review offers a comprehensive and timely overview of dysregulated non-coding RNAs (miRNAs, lncRNAs, circRNAs) in uterine fibroid pathogenesis. It effectively integrates genetic, hormonal and racial factors influencing ncRNA expression and links them to key pathological features such as proliferation, ECM remodeling, and inflammation. The manuscript is well-referenced and clinically relevant. Overall, with minor edits the paper is a valuable contribution to fibroid molecular research. Following minor revisions are needed:
- The term "Fibs" is overused and informal. Use the full term "fibroids" or "uterine fibroids" throughout.
- Line 13 “The growth of these tumors are driven by…” instead of “are driven” write “is driven”.
- The reference to specific miRNAs in other cancers (e.g., colon, breast) is informative but should clearly tie back to fibroids to maintain focus.
- Line 91: “Recent evidence is emerging implicating circRNAs…”. Rewrite as "Recent studies implicate circRNAs..."
- Line 179 “constructing a complete microRNAome profile...". Use "comprehensive" or "global" for better scientific tone.
- Line 189 “down regulation” should be “downregulation” (one word)
- Lines 202–214 reiterate similar outcomes (apoptosis suppression, ECM remodeling). Combine the apoptotic and ECM-modulating effects into a cohesive narrative: "miR-21a-5p promotes fibroid cell survival by concurrently suppressing apoptosis via PCD-4 inhibition and enhancing ECM remodeling through upregulation of TGF-β3 and MMP-2/11."
- Line 206, PCD 4 should be PDCD 4 which is a standard abbreviation.
- Line 218: "are pivotal in Fib pathogenesis". Rephrase the sentence as “play a crucial role in fibroid pathogenesis…”
- Line 540 “missexpression” is typo. Revise it to dysregulated expression" or "misexpression."
- Line 537: Typo in “progesterone”. Should be “progesterone”

Author Response
Thanks for your suggestions. We edited our manuscript as suggested.
Reviewer 2 Report
Comments and Suggestions for Authors
Uterine fibroids are benign tumors that affect ~70% of women. The growth of these tumors are driven by estrogen and progesterone.The study of genetics related to the most common benign pathology in fertile women. The conclusions are important in providing the scientific community with data to support future research in the therapeutic and preventive fields, such as screening populations with high genetic risk with implications for fertility, clearly in favor of pharmaceutical research. I would say that the work is well conducted, extremely complete and comprehensive, useful in the field of gynecology to answer questions relating to a pathogenesis that is still poorly understood and that, with the current methods under development for the identification and deletion of gene sequences causing fibroids, could open doors to the therapeutic field (as therapeutic targeting of these dysregulated ncRNAs for amore precise and individualized non-hormonal based treatment ). The work is publishable.
Author Response
We appreciate your supportive comments.
Reviewer 3 Report
Comments and Suggestions for Authors I really appreciate the work you have done in writing this manuscript. I believe it makes aconsiderable contribution to the understanding of leiomyomas, their prevention and treatment.
However, I suggest that you create some explanatory diagrams for the miRs described, so the text
will be easier to read.
Author Response
We appreciate your supportive comments.
Reviewer 4 Report
Comments and Suggestions for Authors
In "The Role of Non-coding RNAs in the Pathogenesis of Uterine Fibroids" by Boos and colleagues, the authors aim at a comprehensive literature review on this topic.
In general, this is a well-written, comprehensive review, of interest to scientists and clinicials in this oncology field.
I have several important suggestions, mainly to make the review even more attractive to readers:
For part 1: General discussion of lncRNA's:
- Are all lncRNA classes described? Please mention.
- This section would benefit from a large Figure showing origin, length, metabolism, intracellular location, function etc of the different lncRNA's.
For part 2: lncRNA's in UF:
- The authors should describe their literature search strategy, and/or describe how they identified the profiling studies in e.g. the beginning of the miR section.
- It would be very informative if the authors do not just discuss the most over-expressed lncRNA's from the different studies, but also show in a Table what the differences/similarities in the top lists were between the different studies. This will provide a clearer overview of the studies.
- Can the authors structure the tables with additional columns showing e.g.: type of experiment, in vitro/in vivo, technology, etc.
- Would it be possible to add one or more Figures, maybe per large lncRNA class, as for Section 1? Connections between the different lncRNA's and subcellular location/cellular processes (as mentioned briefly in the Conclusion) can be visualized by this (e.g. COL genes (ECM, fibrosis) with miR-29, miR-93 and miR-139-5p. Similarly for H19/MIAT).
Author Response
Comments and Suggestions for Authors
In "The Role of Non-coding RNAs in the Pathogenesis of Uterine Fibroids" by Boos and colleagues, the authors aim at a comprehensive literature review on this topic.
In general, this is a well-written, comprehensive review, of interest to scientists and clinicials in this oncology field.
I have several important suggestions, mainly to make the review even more attractive to readers:
For part 1: General discussion of lncRNA's:
- Are all lncRNA classes described? Please mention.
Response: Thanks for your suggestion. Yes, all lncRNA classes are described (line 69-97).
- This section would benefit from a large Figure showing origin, length, metabolism, intracellular location, function etc of the different lncRNA's.
Response: We appreciate the reviewer’s suggestion to include a comprehensive figure summarizing the origin, length, metabolism, intracellular localization, and function of the lncRNAs discussed. While our current description aims to provide a clear and concise narrative, we agree that such a figure could serve as a valuable visual aid for readers. Given the complexity and diversity of lncRNAs in fibroids, a single figure may not capture all these dimensions without becoming overly dense. We will make sure to summary the most relevant features of the key lncRNAs discussed in the table 2.
For part 2: lncRNA's in UF:
- The authors should describe their literature search strategy, and/or describe how they identified the profiling studies in e.g. the beginning of the miR section.
Response: Thanks for your suggestion. We added the below sentences in the beginning of the miR section.
“We conducted a comprehensive literature search in PubMed, Scopus, and Google Scholar to identify studies on non-coding RNAs in uterine fibroids. Search terms included combinations of keywords such as long non-coding RNA, lncRNA, microRNA, miRNA, fibroid, leiomyoma, gene expression profiling, and RNA sequencing. Relevant studies were further identified by screening the reference lists of retrieved articles.”
- It would be very informative if the authors do not just discuss the most over-expressed lncRNA's from the different studies, but also show in a Table what the differences/similarities in the top lists were between the different studies. This will provide a clearer overview of the studies.
Response: We thank the reviewer for the valuable suggestion to include a comparative table summarizing the similarities and differences among the top overexpressed lncRNAs reported in various studies. While we agree that such a table could provide a useful overview, RNA-seq findings generally require further validation by qRT-PCR, and most of the published studies do not provide such confirmatory data. Therefore, our current narrative was designed to highlight the key findings from each dataset individually.
- Can the authors structure the tables with additional columns showing e.g.: type of experiment, in vitro/in vivo, technology, etc.
Response: We appreciate the reviewer’s suggestion to include additional columns in the tables indicating details such as type of experiment, in vitro/in vivo context, and technology used. Our current table format was designed to be concise and focused on the key findings without making the tables overly complex.
- Would it be possible to add one or more Figures, maybe per large lncRNA class, as for Section 1? Connections between the different lncRNA's and subcellular location/cellular processes (as mentioned briefly in the Conclusion) can be visualized by this (e.g. COL genes (ECM, fibrosis) with miR-29, miR-93 and miR-139-5p. Similarly for H19/MIAT).
Response: We thank the reviewer for the thoughtful suggestion to include additional figures illustrating the connections between major lncRNA classes, their subcellular localization, and associated cellular processes. We agree that such visualizations could help readers integrate these relationships; however, the subcellular location and interaction of lncRNA with miRNA and downstream genes have not been determined to date.
